# Pain assessment in horses using automatic facial expression recognition through deep learning-based modeling

Gabriel Carreira Lencioni[1]*, Rafael Vieira de Sousa[2], Edson José de Souza Sardinha[2], Rodrigo Romero Corrêa[3], Adroaldo José Zanella[1]

1 Department of Preventive Veterinary Medicine and Animal Health of the School of Veterinary Medicine and Animal Science (FMVZ) of the University of São Paulo (USP), São Paulo, SP, Brazil, 2 Department of Biosystems Engineering, Faculty of Animal Science and Food Engineering (FZEA), of the University of São Paulo, Pirassununga, São Paulo, Brazil, 3 Department of Surgery of the School of Veterinary Medicine and Animal Science (FMVZ) of the University of São Paulo (USP), São Paulo, SP, Brazil

* gabriel.lencioni@usp.br

**Data Availability Statement:** All data are available in the Mendeley repository under: Lencioni, Gabriel; Sousa, Rafael; Sardinha, Edson; Romero, Rodrigo; Zanella, Adroaldo (2021), "Automatic Pain

## Abstract

The aim of this study was to develop and evaluate a machine vision algorithm to assess the pain level in horses, using an automatic computational classifier based on the Horse Grimace Scale (HGS) and trained by machine learning method. The use of the Horse Grimace Scale is dependent on a human observer, who most of the time does not have availability to evaluate the animal for long periods and must also be well trained in order to apply the evaluation system correctly. In addition, even with adequate training, the presence of an unknown person near an animal in pain can result in behavioral changes, making the evaluation more complex. As a possible solution, the automatic video-imaging system will be able to monitor pain responses in horses more accurately and in real-time, and thus allow an earlier diagnosis and more efficient treatment for the affected animals. This study is based on assessment of facial expressions of 7 horses that underwent castration, collected through a video system positioned on the top of the feeder station, capturing images at 4 distinct timepoints daily for two days before and four days after surgical castration. A labeling process was applied to build a pain facial image database and machine learning methods were used to train the computational pain classifier. The machine vision algorithm was developed through the training of a Convolutional Neural Network (CNN) that resulted in an overall accuracy of 75.8% while classifying pain on three levels: not present, moderately present, and obviously present. While classifying between two categories (pain not present and pain present) the overall accuracy reached 88.3%. Although there are some improvements to be made in order to use the system in a daily routine, the model appears promising and capable of measuring pain on images of horses automatically through facial expressions, collected from video images.

Assessment in Horses", Mendeley Data, V3, doi:
10.17632/t8rtzcgwxm.3

**Funding:** Gabriel Carreira Lencioni received an
undergraduate stipend from the University of São
Paulo, during the course of the experimental work.
Brazilian Coordination for the Improvement of
Higher Education (CAPES/PROEX - 760/2020) has
supported this study with the publication fees. The
funders had no role in study design, data collection
and analysis, decision to publish, or preparation of
the manuscript.

**Competing interests:** The authors have declared
that no competing interests exist.

## Introduction

Recognizing pain correctly in animals is essential in order to guarantee their welfare and to
provide successful and rapid treatment when needed [1, 2]. Untreated pain severely compro-
mises the welfare of horses and also can have long-term, permanent consequences: for exam-
ple, sensitizing the central nervous system, altering the threshold for responses, subsequently
causing hyperalgesia among other problems [3]. It is suggested that pain evaluation should be
seen as the fifth vital sign and, therefore, should be monitored as often as heart rate, respiratory
rate, blood pressure, and body temperature [4], in order to preserve the health and welfare of
the animal. A continuous, real-time evaluation of pain expression over time will result in better
analgesic treatment and recovery [4]. The requirement to evaluate pain many times a day
needs to be balanced in a way that the procedure of the evaluation itself does not result in stress
levels as a consequence of repeated physical interventions, which could have a negative impact
on the patient's overall wellbeing [4].

Evaluating pain correctly can be a challenging task. It requires ability and training from the
observer in order to detect pain-related changes in behavioral or physiological parameters of
the animal [4]. In addition, it is extremely difficult to apply these evaluations to the daily rou-
tine of a hospital or equine center, since it requires experienced observers and prolonged
observation periods [2, 4].

Facial cues are used in order to assess pain and other emotional states in humans, mostly
on patients that are unable to verbalize to their doctors what they are feeling, such as infants
and patients with cognitive impairment, for example [2, 5, 6].

The study of facial expressions initially proposed by Charles Darwin showed that there are
similarities in the facial expressions of humans and non-human animals [7]. Beyond that, it
has been shown recently that humans have the ability to recognize emotions in several animal
species, including pain by analysis of facial expressions [8, 9].

The systematic use of facial expressions as a tool to assess pain in non-human animals was
initially proposed in 2010 when a mouse grimace scale was developed [10]. Protocols to assess
facial expressions have been published for several species, including horses [2, 4, 11, 12]. A
recent study has also analyzed the use of the Equine Facial Action Coding System (EquiFACS)
[13] and concluded that facial expressions can be a reliable indicator of pain in horses and also
found that facial expressions related to pain, described by EquiFACS, occur in the same ana-
tomical regions as described by the Horse Grimace Scale (HGS) [2] and An Equine Pain Face
protocol [11, 14]. It is important to mention that, in order to guarantee the right application of
pain recognition protocols, observers must be well trained to recognize these facial cues in
order to minimize individual bias and also be available at different times of the day to make
consistent evaluations, which requires significant amounts of time [15, 16]. In addition, since
horses are classified from the evolutionary perspective as prey animals, it is possible that pain
behaviors end up being suppressed due to the responses to potential threatening stimuli, such
as the presence of an unfamiliar observer [2, 4, 11, 15], and certain pain behaviors will proba-
bly be displayed only in the absence of human observers [15].

The use of automated systems in veterinary practice, animal behavior assessment, and
breeding systems has increased, with emphasis on, among others, the use of machine vision
that associates image capture sensors with algorithms using Artificial Intelligence methods,
such as those belonging to the Machine Learning framework tools [17–19]. Due to the chal-
lenges regarding pain evaluation, machine learning methods for pain assessment are promis-
ing since it offers noninvasive full-time surveillance without stressing animals, making it
possible to carry out continuous analysis of the evolution of treatment or even as a tool for
early diagnosis by sending warning signals to the veterinarian instantaneously. Automated

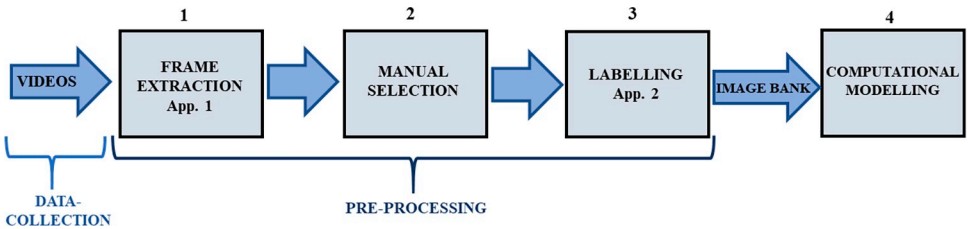

**Fig 1. Methodology steps for building the machine vision algorithm for pain assessment in horses.**

pain assessment has been evaluated in many animal species such as mice, sheep, and even humans [15, 20, 21]. A recent article addressed the complexities of assessing pain on horses through automatic recognition systems [22]. In addition, this resource could also be very useful as a way to educate students using a more visual and practical approach to pain recognition in horses [15].

Given this background, the aim of this study is to develop a computational automated classifier capable of detecting pain in horses using cameras in order to capture and evaluate their facial expressions based on the Horse Grimace Scale [2].

## Materials and methods

The methodology followed the steps presented in the flowchart (Fig 1). Seven horses undergoing routine castration were filmed two days before and four days after the procedure. These videos were then processed by an application software (App. 1) developed using the software Matlab 2016b (Mathworks, USA) in order to extract frames that were then manually selected, looking for images that allowed the ideal view to evaluate pain using facial expressions. These images were analyzed and labeled by an expert using the HGS [2], supported by an application software (App. 2) developed using the software Matlab 2016b. The labeling process enabled the organization of the image database composed of images of horses expressing distinct levels of pain: pain not present, moderately present, and obviously present. This image database was used for the training of the computational algorithm classifier (computational modeling).

### Data-collection

Video images were collected from seven horses of approximately one year of age (six of the breed Brazilian sport horse and one Mangalarga Marchador, another Brazilian breed) that underwent routine surgical castration, for management reasons, unrelated to the current experiment. Castration was requested by the University of São Paulo, owner of the animals, that gave to the experimenter's permission to collect video data. The surgical procedure followed the standard protocol for sedation, analgesia, anesthesia and pain management carried out at the veterinary hospital of the School of Veterinary Medicine and Animal Science of the University of São Paulo [23]. The horses were sedated using intravenous xylazine (0.8 mg/kg) and intramuscular morphine (0.1 mg/kg) for analgesia. The anesthetic induction was done using a combination of ketamine (2 mg/kg) and diazepam (0.05 mg/kg) through intravenous administration. The anesthetic maintenance was performed by an association of glyceryl guaiacol ether (50mg/ml), ketamine (2mg/ml), and xylazine (0.5 mg/ml) by slow intravenous infusion. During maintenance of the anesthesia, heart and respiratory rate were evaluated every 5 minutes. The presence of palpebral reflex, the eye position and nystagmus were evaluated continuously in order to verify that the animal was at an adequate level of anesthesia. Local anesthesia was administered in two lines parallel to the scrotal median raphe, using 10

ml of 2% lidocaine in each one. Intratesticular local anesthesia was also administered using the same drug, with 5 ml for each testicle. Orchiectomy was performed using the closed technique; a skin incision of approximately eight centimeters was made parallel to the scrotal median raphe, in the most ventral region of the scrotum. Skin and dartos tunic were incised and the testicle was exteriorized still covered by the vaginal tunics. The emasculator was positioned and compressed on the spermatic cord and maintained for five minutes. The spermatic funiculus was incised and the testicle, covered by the tunics, was removed. This procedure was repeated in the same way for the contralateral testicle. The total length of the procedure was approximately 40 minutes, including 20 minutes for anesthesia and about 20 minutes for the surgery procedure itself.

The animals were monitored side by side inside the feeder station using a camera system positioned in front of the feeder, for two days before and four days after the procedure, at four distinct time points of the day: 7 am, 10 am, 12 pm and 4 pm, aiming to capture images of the animals while presenting distinct levels of pain. The images collected resulted in 320 videos of 30 minutes each, acquired using Intelbras VHD 1220 B–G4 Multi HD cameras.

All the animals were treated with the same postoperative treatment, which included systemic administration of benzathine penicillin (40,000 IU/Kg intramuscularly and in a single dose), flunixin meglumine (1.1mg/kg intravenously, once daily for 3 days), and anti-tetanus serum (10,000 IU intramuscularly, in a single dose). The post-castration follow ups were carried out in the morning (7 am) and in the afternoon (12 pm), before the video recordings. The scrotum wounds were washed with soap and water and treated with penicillin-based ointment twice a day. The protocol was reviewed and approved by the Ethical Committee on the use of animals (CEUA number: 6603170419). All the animals filmed were property of the University of Sao Paulo and the written consent for this study was obtained.

## Pre-processing and organization of image database

The videos were processed through App. 1 in order to automatically detect motion and extract frames (images) of each horse at different moments, resulting in 185672 images. Then, 3000 images were selected manually by visual inspection in order to use only the ones that were in the right position, with a lateral view of the horse's head, in order to evaluate pain through facial expression [2].

The HGS is composed of six parameters:1- Position of the ears, 2- Orbital tightening, 3- Tension above the eye area, 4- Prominent strained chewing muscles, 5- Mouth strained and pronounced chin, 6- Strained nostrils and flattening of the profile. These parameters allow for the evaluation of three levels of pain: not present, moderately present and obviously present [2]. In order to test the viability of the system, those six parameters were grouped as three different groups of face parts (parameters): the first one was regarding the position of the ears; the second one regarding the eye (such as orbital tightening and tension above eye area); and the third grouped the chewing muscles and mouth and nostrils position. Fig 2 shows the three parts of the horse's face and the three pain categories that are used in modeling the machine vision algorithm.

The 3000 images were identified, cropped, and labeled by a trained observer based on the HGS [2] and using App. 2 (Fig 3) to compose a final image database of three different pain categories for each one of the regions of the face.

The final database was composed of 4850 images having 2379 images of ears, 1436 images of eyes, and 1035 images of mouth and nostrils, all classified according to the three defined categories (Table 1). These images originated from the 3000 selected frames which were cropped into specific areas related to the parameters used on the HGS [2].

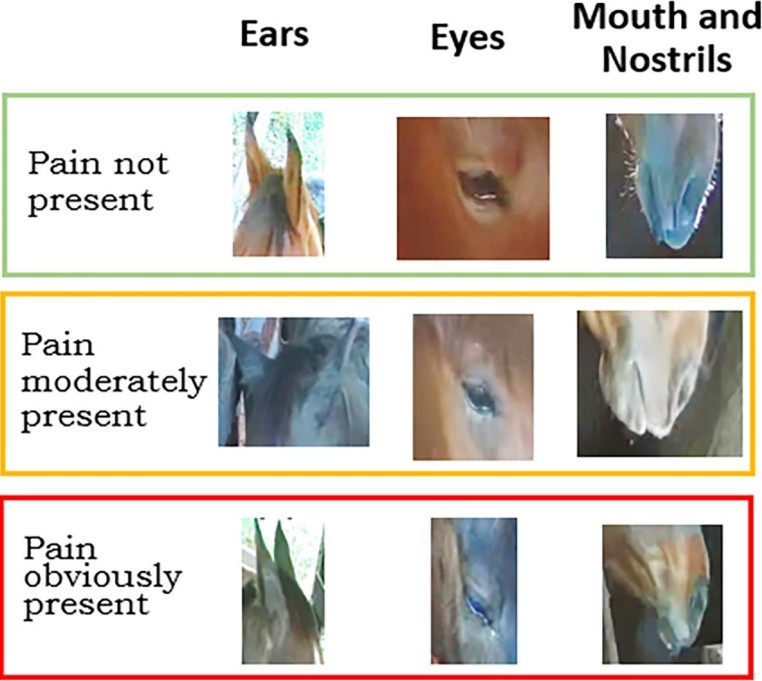

**Fig 2. Horse face regions and pain levels for evaluation by computer modeling.** Ears: Considers ear position, distance between them and direction. Eyes: Considers size of the orbital area, muscular tension above the eye, visibility of underlying bone surfaces. Mouth and nostrils: Considers level of mouth and nostrils straining, tension, size of mouth-collum (line between the upper and lower lips) and nostril dilation.

## Computational modeling

After the image database was organized, three pain classifier models based on Convolutional Neural Network (CNN) architecture were built according to the Sequential Keras Library (version 2.0.6) for Python Programming Language (version 3.5.4 rc1) and set up (Table 2) for each one of the three parameters (ears, eyes, and mouth and nostrils). The image database was used as a predictive attribute (model input) and the pain level as a target attribute (model output). For each of the three classifier models, the modeling cycle was accomplished using 80% of images for the training itself, 10% for validation (used during the training process to assess the error and establish a process stop condition), and 10% for testing (used at the end of the modeling cycle to evaluate the performance of the pain prediction model). Images were randomly assigned for training, validation and testing steps. Several cycles of the CNN architecture were executed and adjusted for determining the best configuration of the hyperparameters (Table 3) that produced the best pain model classifier for each target attribute. At end of each modeling cycle, the model´s performance was evaluated using confusion matrix metrics (accuracy, precision, and recall).

The accuracy indicates the overall efficacy of the model and is calculated by:

$$Accuracy = (True\ positives + True\ negatives)/Total$$

The precision indicates whether the data was classified in the correct class or not, and is calculated by:

$$Precision = True\ positives/(True\ positives + False\ positives)$$

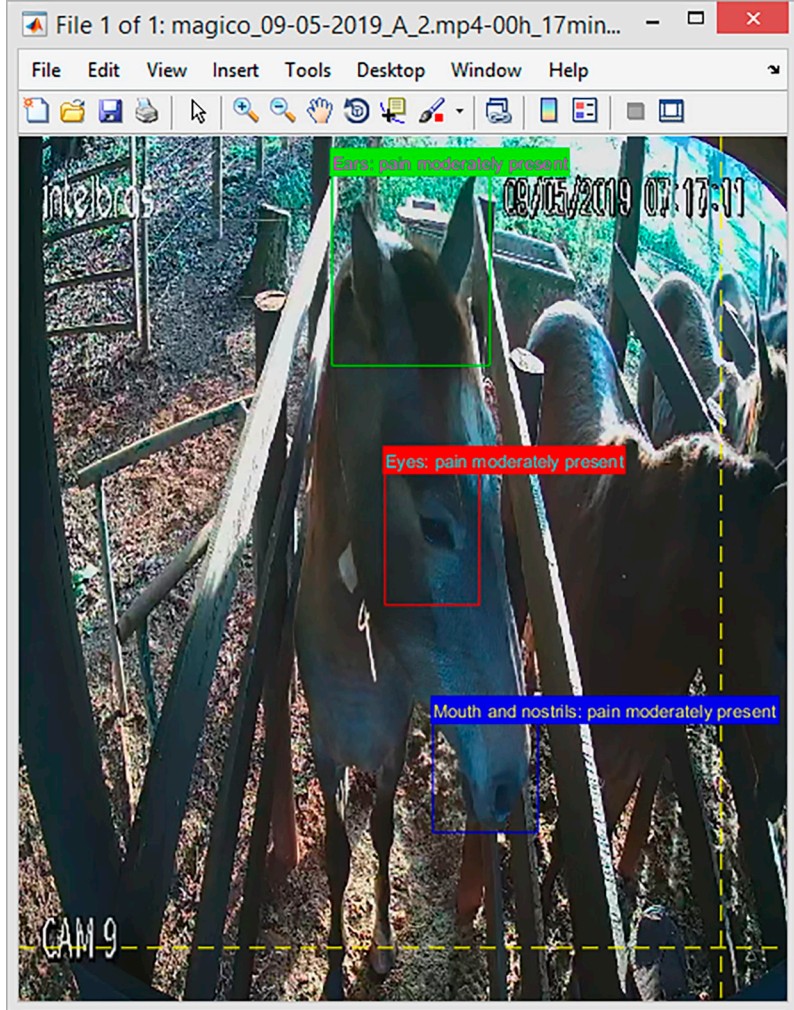

**Fig 3. Window application software for the labeling process.**

Recall measures the ability of the classifier to identify all the correct data for each class and is calculated by:

$$Recall = True\ positives/(True\ positives + False\ Negatives)$$

Afterwards, each one of the final trained models was used together through a classification process in order to classify pain on images showing the whole face of the animals based on all the recommendations and confidence values. The classifier was created using a machine

**Table 1. Composition of final image database.**

| Class | Number of images | | |
|---|---|---|---|
| | **Ears** | **Eye** | **Mouth and Nostrils** |
| Pain not present | 1991 | 477 | 506 |
| Pain moderately present | 203 | 488 | 393 |
| Pain obviously present | 185 | 471 | 136 |
| **Total** | **2379** | **1436** | **1035** |

**Table 2. Model architecture based on the convolutional neural network that was used to classify the pain level for each evaluated parameter (ears, eye and mouth and nostrils).**

| Layer | Configuration |
|---|---|
| 2D Convolutional | 60 kernels with size = 7x7; activation by Rectified Linear Unit |
| 2D MaxPooling | Pooling size = 2x2 |
| Batch Normalization | Default |
| 2D Convolutional | 32 kernels with size = 5x5; activation by Rectified Linear Unit |
| 2D MaxPooling | Pooling size = 2x2 |
| Batch Normalization | Default |
| Flattening | Default |
| Fully Connected | 500 or 1000 Neurons (See Table 3); activation by Rectified Linear Unit |
| Fully Connected | 3 Neurons; activation by Softmax |

learning method, based on Artificial Neural Network (ANN-based classifier). The ANN-based classifier was built using the Weka 3.8 software (Waikato Environment for Knowledge Analysis, New Zealand) and a Perceptron feedforward and multi-layered architecture, with a sigmoid transfer function in the hidden layer and a linear transfer function in the output layer. The Levenberg-Marquardt backpropagation method and mean squared error were employed to measure the performance using k-fold cross validation of 10. ANN-based classifier performance indicators (accuracy, precision and recall) are calculated by averaging the values among all folds.

Forty complete original images were randomly selected of animals from each class: no pain, moderately present pain and obviously present pain, resulting in 120 complete images. A total of 360 cuttings from the regions of these samples were used to create a database that was used in the process of training and testing to obtain the final ANN-based classifier.

## Results and discussion

The CNN-based individual training models created for each part of the face resulted in an overall accuracy of 90.3% for the ears (Table 4), 65.5% for the eyes (Table 5), and 74.5% for the mouth and nostrils (Table 6).

The model trained to evaluate pain level according to the position of the ears was composed of 2379 images, distributed as 1991 images showing absence of pain, 203 showing pain moderately present and 185 showing pain as being obviously present (Table 1). The generated model based on ears was tested using 237 images from the separated dataset for testing and showed a

**Table 3. Hyperparameters used in the final classifier model for each evaluated parameter (ears, eye, and mouth and nostrils).**

| Hyperparameter | Model | | |
|---|---|---|---|
| | Ears | Eye | Mouth and Nostrils |
| Number of trained models | 3 | 3 | 3 |
| Number of epochs per model | 20 | 30 | 30 |
| Number of neurons in the last but one Fully Connected layer | 1000 | 500 | 1000 |
| Width x Height of input image | 200x200 | 80x80 | 120x120 |
| Number of images used for training (80% of total for each class) | 1905 | 1152 | 831 |
| Number of images used for validation (10% of total for each class) | 237 | 142 | 102 |
| Number of images used for test (10% of total for each class) | 237 | 142 | 102 |
| Batch size | 50 | 50 | 50 |

**Table 4. Confusion matrix between the data classified by the CNN-based model and HGS for the ears images.**

| HGS-Based Classified Pain | CNN-Based Classified Pain | | | Recall |
|---|---|---|---|---|
| | Not Present | Moderately Present | Obviously Present | |
| Not Present | 198 | 0 | 1 | 99.5% |
| Moderately Present | 8 | 11 | 1 | 55.0% |
| Obviously Present | 11 | 2 | 5 | 27.8% |
| Precision | 91.2% | 84.6% | 71.4% | 90.3% |

Gray cells indicate the correct predictions in each class (accuracy). CNN: Convolutional Neural Network. HGS: Horse Grimace Scale.

good ability to identify the absence of pain, but it did not obtain the same performance in the differentiation between the Moderately Present and Obviously Present classes, as can be seen by the lower values of the parameters recall and precision of these classes (Table 4). This fact is reinforced by the lower values of recall and precision (27.8% and 71.4%, respectively) of the class 'Obviously Present' in relation to the other classes. This result indicates the need for better balancing of the image database with more samples in order to get better performance in the training process in future CNN-based modeling works.

The model trained to evaluate the presence of pain when analyzing the eyes, on the other hand, was well balanced, having 1436 images, distributed as 477 images showing absence of pain, 488 moderately present pain, and 471 obviously present pain (Table 1). In this case, the general accuracy of the model of 65.5% (Table 5), when tested using 142 images from the separated dataset for testing, was lower than the 90.3% accuracy of the pain assessment model by the position of the ears (Table 4) mainly due to the values of False Positive of the class Moderately Present and Obviously Present. In general, the smaller number of samples used in training in relation to the model for indication of pain in the ears (1463 eyes images against 2379 ears images) may explain this inferior performance.

The database of the third model referring to the mouth and nostrils is the smallest one, but it has an intermediate balance of images in relation to the images of the ears and the eyes. The 1035 images are distributed in 506 images of the absence of pain, 393 of moderately present pain, and 136 of obviously present pain (Table 1). With this structure, it resulted in an intermediate accuracy of 74.5% (Table 6), when tested using 102 images from the separated dataset for testing, relative to the models for the ear and the eye, which had accuracies of 90.3% (Table 4) and 65.5% (Table 5), respectively.

From the three ready classification algorithms, a fine-tuning process was performed to train the ANN-based classifier that combines the individual recommendations and confidence value of each model. The final value for the number of neurons in the hidden layer, the learning rate and the momentum hyperparameters were 5, 0.3 and 0.2 respectively.

**Table 5. Confusion matrix between the data classified by the CNN-based model and HGS for the eyes images.**

| HGS-Based Classified Pain | CNN-Based Classified Pain | | | Recall |
|---|---|---|---|---|
| | Not Present | Moderately Present | Obviously Present | |
| Not Present | 38 | 2 | 7 | 80.9% |
| Moderately Present | 16 | 27 | 5 | 56.3% |
| Obviously Present | 8 | 11 | 28 | 59.6% |
| Precision | 61.3% | 67.5% | 70.0% | 65.5% |

Gray cells indicate the correct predictions in each class (accuracy). CNN: Convolutional Neural Network. HGS: Horse Grimace Scale.

**Table 6. Confusion matrix between the data classified by the CNN-based model and HGS for the mouth and nostrils images.**

| HGS-Based Classified Pain | CNN-Based Classified Pain | | | Recall |
|---|---|---|---|---|
| | Not Present | Moderately Present | Obviously Present | |
| Not Present | 44 | 4 | 2 | 88.0% |
| Moderately Present | 11 | 26 | 2 | 66.7% |
| Obviously Present | 3 | 4 | 6 | 46.2% |
| Precision | 75.9% | 76.5% | 60.0% | 74.5% |

Gray cells indicate the correct predictions in each class (accuracy). CNN: Convolutional Neural Network. HGS: Horse Grimace Scale.

While evaluating the pain level on the complete image of the horses, the system was able to achieve an overall accuracy of 75.8%. When analyzing the confusion matrix (Table 7) between each one of the classes, it showed that 80% of the time that the system predicted that the horse on the image was presenting absence of pain, the animal was indeed showing this classification according to the human observer that had evaluated the images previously. It is interesting to note that the 17.5% error made by the system in this class was classified as pain moderately present, and only 2.5% as pain obviously present, meaning that the mistake was close to the correct evaluation. For the next class, when the system predicted that the animals presented moderate pain, it was right in 67.5% of the cases, with the error distributed between pain not present (17.5%), and pain obviously present (15%). In the third class, 80% of the time that the system predicted that the horse was clearly in pain, this was precise according to the human evaluator. It is also interesting that the 20% of error was classified as moderately present pain, but not as pain not present, suggesting that even the mistakes of the system show a good level of learning from the parameters.

When evaluating the results of the automated system while differentiating between two categories (pain not present and present pain) (Table 8) instead of three levels (pain not present, moderately present, and obviously present) the outcome was even more promising, reaching an overall accuracy of 88.3%.

These results show that evaluating pain automatically in horses through the use of artificial intelligence to recognize facial expressions is very promising. However, some improvements should be made in the future in order to produce even better outcomes, such as having a larger image bank with higher quality images in order to improve the training of the Convolutional Neural Networks, and also for a better balance between each one of the classes, resulting in an equivalent number of images for each class. Additionally, it would be interesting to have more trained evaluators classifying the images in order to exclude possible biases from individual discrepancies when applying the pain scales. It is also important to train the system to recognize other behaviors presented by the horses that could lead to an incorrect interpretation of pain.

**Table 7. Confusion matrix between the data classified by the ANN-based classifier and HGS, described on three levels of pain.**

| HGS-Based Classified Pain | ANN-Based Classified Pain | | | Recall |
|---|---|---|---|---|
| | Not Present | Moderately Present | Obviously Present | |
| Not Present | 32 | 7 | 1 | 80.0% |
| Moderately Present | 7 | 27 | 6 | 67.5% |
| Obviously Present | 0 | 8 | 32 | 80.0% |
| Precision | 82.1% | 64.3% | 82.1% | 75.8% |

Gray cells indicate the correct predictions in each class (accuracy). ANN: Artificial Neural Network. HGS: Horse Grimace Scale.

**Table 8. Confusion matrix between the data classified by the ANN-based classifier and HGS, described on two levels of pain.**

| HGS-Based Classified Pain | ANN-Based Classified Pain | | Recall |
|---|---|---|---|
| | **Not Present** | **Present** | |
| Not Present | 32 | 8 | 80.0% |
| Present | 6 | 74 | 92.5% |
| Precision | 84.2% | 90.2% | 88.3% |

Gray cells indicate the correct predictions in each class (accuracy). ANN: Artificial Neural Network. HGS: Horse Grimace Scale.

## Conclusions

The model based on Convolutional Neural Network showed potential to assess the level of pain in horses. The automated computational classifier showed a success rate of 75.8% when evaluating three levels of pain in horses and 88.3% when discriminating the presence or not of pain through the analysis of the animals' facial expressions. Even with the high performance for detecting pain levels, we found that it is still necessary to carry out new measurements to expand the database in relation to the categories, especially for the purpose of balancing them. In addition, by increasing the quality of the images used, it will be possible to better identify the expressions of pain and to generate a more accurate labeling of the images according to the levels of pain for the training of models. This study seeks to show the potential of the proposed method. Our aim is to inspire other scientists who can facilitate the improvement of networks of excellence to collaborate on the development of intelligent systems to assess pain in animals.

## Acknowledgments

We acknowledge the help from the Department of Preventive Veterinary Medicine and Animal Health, and the Department of Surgery, both from the School of Veterinary Medicine and Animal Science. The Horse Sector of the Campus Fernando Costa is also acknowledged for the help with the management of the animals. The veterinarian Laura Pinseta helped with the required exams of the animals. Leandro Sabei, M.Sc., Thiago Bernardino, M.Sc., and Denis Sato, M.Sc. helped to install the video system for data collection. Erica N. Feuerbacher helped with the revision process. We would also like to show our gratitude to the team from the CAEP/FMVZ, the members from CECSBE-USP, and the Innovation and Entrepreneurship Program–InovaGrad-USP, from AUSPIN and PRG-USP.

## Author Contributions

**Conceptualization:** Adroaldo José Zanella.

**Data curation:** Gabriel Carreira Lencioni.

**Funding acquisition:** Adroaldo José Zanella.

**Methodology:** Gabriel Carreira Lencioni, Rafael Vieira de Sousa, Edson José de Souza Sardinha, Rodrigo Romero Corrêa.

**Project administration:** Gabriel Carreira Lencioni, Adroaldo José Zanella.

**Software:** Rafael Vieira de Sousa, Edson José de Souza Sardinha.

**Supervision:** Adroaldo José Zanella.

**Writing – original draft:** Gabriel Carreira Lencioni.

**Writing – review & editing:** Rafael Vieira de Sousa, Edson José de Souza Sardinha, Rodrigo Romero Corrêa, Adroaldo José Zanella.

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
