## [Decision Letter · Decision Letter 0]

13 May 2021

PONE-D-21-11680

Pain assessment in horses using automatic facial recognition through deep learning-based modeling

PLOS ONE

Dear Dr. Lencioni,

Thank you for submitting your manuscript to PLOS ONE. After careful consideration, we feel that it has merit but does not fully meet PLOS ONE’s publication criteria as it currently stands. Therefore, we invite you to submit a revised version of the manuscript that addresses the points raised during the review process.

 Please submit your revised manuscript by Jun 27 2021 11:59PM. If you will need more time than this to complete your revisions, please reply to this message or contact the journal office at plosone@plos.org. Please include the following items when submitting your revised manuscript:

We look forward to receiving your revised manuscript.

Kind regards,

Humaira Nisar

Academic Editor

PLOS ONE

Journal Requirements:

2. In your Methods section, please provide additional details regarding participant consent from the owners of the animals. In the ethics statement in the Methods and online submission information, please ensure that you have specified (1) whether consent was informed and (2) what type you obtained (for instance, written or verbal). If the need for consent was waived by the ethics committee, please include this information.

3.In your Data Availability statement, you have not specified where the minimal data set underlying the results described in your manuscript can be found. PLOS defines a study's minimal data set as the underlying data used to reach the conclusions drawn in the manuscript and any additional data required to replicate the reported study findings in their entirety. All PLOS journals require that the minimal data set be made fully available. For more information about our data policy, please see http://journals.plos.org/plosone/s/data-availability.

4.Thank you for stating the following financial disclosure:

5. Please upload a copy of Figure 5, to which you refer in your text on page 18. If the figure is no longer to be included as part of the submission please remove all reference to it within the text.

Additional Editor Comments:

The manuscript aims to develop an automated method for the detection of post-operation pain of horses using convolutional neural networks.

The topic and the work is very interesting.

However reviewers have major concerns about the technical aspects of the paper, as many important parameters have not been explained appropriately. Which is probably because of the reason that the paper writing also needs a major improvement.

The reviewers have given detailed comments which will be very useful in improving the quality of this manuscript. Hence based on the reviewers recommendation i will recommend a major revision.

Reviewers' comments:

Reviewer's Responses to Questions

**Comments to the Author**

1. Is the manuscript technically sound, and do the data support the conclusions?

Reviewer #1: Partly

Reviewer #2: Partly

Reviewer #3: Partly

2. Has the statistical analysis been performed appropriately and rigorously? 

Reviewer #1: Yes

Reviewer #2: No

Reviewer #3: N/A

3. Have the authors made all data underlying the findings in their manuscript fully available?

Reviewer #1: Yes

Reviewer #2: No

Reviewer #3: Yes

4. Is the manuscript presented in an intelligible fashion and written in standard English?

Reviewer #1: Yes

Reviewer #2: No

Reviewer #3: No

5. Review Comments to the Author

Reviewer #1: 0.

The manuscript presents and discusses the practical applicability of CNN for the automated detection of post-operation(castration) pain of horses. The work presented involved the design of a shallow 2-conv-layer CNN for the classification of manually cropped images of horse faces, extracted from videos recorded before and after the castration surgery. The importance and relevance of the work are properly outlined in the introduction. The work reported a significant performance enhancement with the ensemble prediction using voting system, compared to individual CNN trained on a single face region of interest (ROI).

Nevertheless, the manuscript seems written in a hurry. Expression of sentences were not properly proof-read by the authors prior to submission.

For example, in Lines 101-108 there are duplication and badly organized phrases. Lines 133-137 are of different font than the other paragraphs.

However, there could still be valuable research contributions with following questions addressed, which have arised in the review process. In addition, the HGS-labelled image dataset generated in this study could potentially chip in to enhance the variation of publicly available HGS dataset for research.

Question 1:

Lines 126-130:

How do 7 horses, for (2+4) days, for 4 times a day, result in 1252 videos?

(7 horses x 6 days x 4 times/day = 168 videos) Were multiple recording cameras used for every horse?

Question 2:

It is commendable and helpful for reviewing that the authors have clearly provided the details like the CNN architecture, CNN kernel size, and the width and height of input images in Table 2 and Table 3.

Could the author explain why shallow CNN of only 2 convolutional layers were used? Couldn't a few additional convolutional layers benefit the feature extraction and abstraction capability of the model as the input are raw images and the output demanded is a highly abstract emotional expression of animal?

Question 3:

It is again commendable that authors honestly and clearly reported the low Recall performance for the presence of pain (esp. for the obvious pain) in each facial subcategory. This is important as the low recall indicates the failed detection of obvious pain which is of greater concern in pain management. This may reflect the insufficience of each individual facial category to serve in pain detection by the presented CNN model.

Has the author not considered increasing the reliability of HGS with independent labelling performed by more than only one trained-observer (as reported in line 159)?

Question 4:

For contents in Lines 202-207 & Lines 252-259:

Based on which image pool was the voting/ensemble model weight-tuned? Was it using the image pool that each member CNN of the ensemble model was trained on (as that presented in Table 1 & 3) or another separate image pool? Or was the weight-tuning also based on the pool of 30 images reported in Table 7 and Table 8 (which shouldn't be the case as the tuning process should not have access to the final test set)? The authors has not explicitly reported the image dataset used for the ensemble/voting weight-tuning process.

If this ensemble voting model with weight ratio of 3:1:1 for eye:ears:mouth&nostril is applied back onto the very original image dataset presented in Table 1 before these images are cropped into ROI, what is the performance of the model is terms of accuracy, precision and recall?

The weight tuning process (which is a key step resulting in the pain classification performance surge) should be analyzed and discussed in greater details.

Reviewer #2: The paper aims at employing artificial intelligence in identifying the existence of pain and its level in horses. Although the problem addressed by the paper is exciting and carries great future potential in opening research avenues, the paper still needs much work to be publication worthy. I have the following concerns:

1. The writing of the paper needs revision. There are many repetitions that should have been omitted after the authors decided on the final shape of the paragraph. The first paragraph of the data collection section (starting line 101) is a prime example. These are not style mistakes but rather genuine writing/editing flaws. See line 58.

2. The authors need to enroll a computer science specialist with knowledge in AI. She/he will contribute to better experiments and to writing in the proper computer science jargon. This is a problem with the current manuscript as it is written by non-specialists.

3. The data collection section, details of the drugs given are not necessary for the contribution of the paper (unless they are required for ethical approval).

4. Section III.II, the number of images 185.000 should be written as 185000, and there is no need for approximately as this is a simple count. The exact number can be easily included.

5. The authors need to elaborate more on the time and effort to inspect 185000 images to select 3000 of them (if not randomly). This does not seem right.

6. The approach to select the proper images could be better than just inspecting the individual frames, which carry a lot of repetitions especially at 30 frames per second video rate.

7. In general, the AI approach need to be driven by a computer scientist so that many of the issues in this paper could be ironed out. For example, the number of subjects (i.e., 7) is too small and may lead to overfitting of the model.

8. The font size in paragraph on line 133 need to be consistent with the paper (such mistakes reflect badly on the quality of the paper).

9. It is not clear how the system combines the result from the three body parts to form a judgement on pain. The current approach seems overly simple.

10. Numbering of sections need to better.

Reviewer #3: The manuscript ‘Pain assessment in horses using automatic facial recognition through deep learning-based modeling’ is an interesting and timely manuscript. The results seem indeed promising to be able to record facial expressions of pain automatically and this is clearly worthwhile for practice.

However, I believe some aspects need to be addressed with more detail and there are a couple of things I do not understand which may need clarification first before the manuscript may be accepted for publication in PLOS ONE.

Below, you will find my arguments for the answers I have given to the questions that are part of the PLOS ONE review process and additional detailed feedback.

Title: I am not sure recognition is the right word to use. To me facial recognition would be that a computer can identify animal 1 from 2 or can tell this is an ear or a mouth. But in this study the assessment was aimed to say that if the ear was positioned like this, there is no pain present. That seems to be something different?

Abstract

Line 2: ‘The aim of the study was ....’ instead of ‘The aim of the study is ....’?

Lines 6 (and 60): Yes, clear that training is important for using the HGS. However, a computer model needs to be trained too and if this is done not properly, it does not really help us forward. So not sure to what extend a computer model is an advantage then?

Lines 8 (and 64-66): True, but a person may not always be needed to be present? The animal can also be monitored from cameras for instance. So this is not always a problem for evaluation, I think.

Line 15: ‘process was applied’ and ‘methods were used’ instead of ‘.... is’ and ‘.... are’? May apply at other sentences too.

Introduction

Line 28: [1,2] instead of [1][2]

Lines 33 and 56: A space is missing before the reference. Please check whether a space is missing at other places perhaps too.

Line 55: The word ‘if’’ is not needed, I think.

Lines 59, 105 and 174: I think here is one space too many. Please check whether an extra space is present at other places perhaps too.

Line 77: Full stop is missing after the references.

Materials and Methods

Lines 88 and 92: I am not sure I understand what the authors mean with application exactly. To be able to reproduce the study, more details on this method may be needed to give.

Lines 101-105: Please check these lines as things are said twice here.

Line 125: What was the total length of the castration procedure? May be good to add to show the reader it was done as quickly as possible.

Line 130: ‘videos of 30 minutes’ instead of ‘videos with 30 minutes’?

Line 130: With 1252 videos, this means you have about (1252 / 7 horses / 6 days / 4 time points =) 7 videos with a total of 3.5 hours of footage around a time point? That seems sufficient, I am just wondering how to visualize this so to say as there is for example only 2 hours in between time point 10 am and 12 pm. When was a video then considered to belong to a certain time point? Or do I not understand this correctly?

Lines 133-137: Seems to be in a different letter font?

Lines 135 and 137: May be good to add when exactly on a day this was done, as I think it should not have interfered with making the videos of Line 130?

Lines 147-148: What is exactly part of the 4th and 5th parameter? You have to guess a bit where the 4th ends and the 5th begins.

Line 157/Fig. 2: I would like to suggest to take another picture as example for the mouth and nostrils parameter for moderately present. As I do not really understand what I am looking at here, I am afraid. If possible, a more clear one as used for not present and obviously present would definitely help.

Lines 159-161: Was the pain assessment solely based on the HGS? Or also based on other behavioral or physiological parameters? In other words, how did the authors know the horse was really in pain? The horses were given post-operative analgesia, so perhaps they did not feel so much pain? For training and testing a computer model this may not be such an issue, but for practice it is. The ears, for instance, could also be turned backwards due to focusing on a sound and not related to pain, and I think that such false warnings to the veterinarian or horse owner is not desired. Or it is the case that only if all three parameters are scored as pain present by the model, a warning is given?

Lines 165-166: I am sorry, but I do not understand how the authors can have a final database of 4847 images if 3000 were selected in the first place? In addition, summing up 2379, 1436 and 1035, I get 4850 images. Somethings seems to be not entirely right here.

Line 173: ‘were built’ instead of ‘was built’

Lines 177-181: I am not an expert on this yet, but what I learned is that others used much more images/frames for training (and testing) the model. Please see for instance: https://arxiv.org/pdf/1909.12605.pdf;
https://www.nature.com/articles/s41598-020-70688-6;
https://www.mdpi.com/1424-8220/19/5/1188/htm Perhaps it depends on what exactly the model needs to be trained in, but it may be good to add here why the authors know/feel these numbers are sufficient.

Line 195/Table 2: Fig. 5 is not included in the manuscript?

Lines 202-203: More details on the voting process would be interesting and relevant to give with respect to be able to reproduce the study.

Lines 205-206: ‘Thus, a variety of ... was tested’ instead of ‘Thus, it was tested a variety of .....’ makes more sense to me.

Results and Discussion

Line 215: I would like to suggest to add ‘obviously’ to this line after ‘being’.

Line 219: It took me some time to figure out where the numbers in Table 4 came from, until I realized that this was based on the test frames which was 237 images for the ears (10% of the total images). It would be good to repeat that somewhere here, I think. Please, be precise and clear which numbers are total and which used for training, validation and testing as the numbers given in Lines 214-215 are the overall total and not the ones used to get Table 4 and that may give a biased picture to the reader. May apply to Lines 231-233, 236-237 and 244-245 as well.

In addition, are there not any results to mention for the validation part of the study?

Line 230: I would like to suggest to rephrase ‘on the eye’ as it feels as strange formulating in this way.

Lines 232 and 245; I would like to suggest to rephrase this as ‘moderately pain present, and xx of obviously pain present’, because that is more clear.

Line 270: I would add ‘clearly’ or ‘obviously’ before ‘in pain’.

Line 286: ‘in horses’ instead of ‘on horses’

Line 289: Indeed a larger databank would be worthwhile. Do the authors have suggestions on the amount of images desired here?

Line 293: Can the authors elaborate a bit more on what this bias would mean if present?

Line 294: Nice to know the authors have in mind to include more behaviours. Do the authors have specific suggestions for these behaviours?

Conclusion

Line 303: Increasing the quality was not mentioned before, or was it? If not, I believe one cannot raise new aspects in the conclusion, so please include the quality in the discussion first as well.

6. PLOS authors have the option to publish the peer review history of their article (what does this mean?). If published, this will include your full peer review and any attached files.

Reviewer #1: No

Reviewer #2: No

Reviewer #3: No

---

## [Author Response · Author response to Decision Letter 0]

26 Jun 2021

Dear Dr. Humaira Nisar and reviewers,

Thank you very much for your comments. They were extremely helpful in order to improve the quality of our manuscript.

In the document below we are describing how we have answered the questions and how we did carry out the necessary corrections to the manuscript.

We have reviewed the manuscript based on the PLOS ONE´s style requirements.

2. In your Methods section, please provide additional details regarding participant consent from the owners of the animals. In the ethics statement in the Methods and online submission information, please ensure that you have specified (1) whether consent was informed and (2) what type you obtained (for instance, written or verbal). If the need for consent was waived by the ethics committee, please include this information.

We have added this information to our Methods section on the manuscript. (Lines 136 – 139).

3.In your Data Availability statement, you have not specified where the minimal data set underlying the results described in your manuscript can be found. PLOS defines a study's minimal data set as the underlying data used to reach the conclusions drawn in the manuscript and any additional data required to replicate the reported study findings in their entirety. All PLOS journals require that the minimal data set be made fully available. For more information about our data policy, please see http://journals.plos.org/plosone/s/data-availability.

Our data are available in the Mendeley Repository: (Lencioni, Gabriel; Sousa, Rafael; Sardinha, Edson; Romero, Rodrigo; Zanella, Adroaldo (2021), “Automatic Pain Assessment in Horses”, Mendeley Data, V1, DOI: 10.17632/t8rtzcgwxm.1).

Our data are available in the Mendeley Repository: (Lencioni, Gabriel; Sousa, Rafael; Sardinha, Edson; Romero, Rodrigo; Zanella, Adroaldo (2021), “Automatic Pain Assessment in Horses”, Mendeley Data, V1, DOI: 10.17632/t8rtzcgwxm.1).

4.Thank you for stating the following financial disclosure:

a. Please clarify the sources of funding (financial or material support) for your study. List the grants or organizations that supported your study, including funding received from your institution.

d. If you did not receive any funding for this study, please state: “The authors received no specific funding for this work.”

5. Please upload a copy of Figure 5, to which you refer in your text on page 18. If the figure is no longer to be included as part of the submission please remove all reference to it within the text.

In the manuscript there is no Figure 5 included. All the references pointing to it on the text were removed.

Additional Editor Comments:

The manuscript aims to develop an automated method for the detection of post-operation pain of horses using convolutional neural networks.

The topic and the work is very interesting.

However reviewers have major concerns about the technical aspects of the paper, as many important parameters have not been explained appropriately. Which is probably because of the reason that the paper writing also needs a major improvement.

The reviewers have given detailed comments which will be very useful in improving the quality of this manuscript. Hence based on the reviewers recommendation i will recommend a major revision.

Comments to the Author

1. Is the manuscript technically sound, and do the data support the conclusions?

Reviewer #1: Partly

Reviewer #2: Partly

Reviewer #3: Partly

2. Has the statistical analysis been performed appropriately and rigorously?

Reviewer #1: Yes

Reviewer #2: No

Reviewer #3: N/A

3. Have the authors made all data underlying the findings in their manuscript fully available?

Reviewer #1: Yes

Reviewer #2: No

Reviewer #3: Yes

4. Is the manuscript presented in an intelligible fashion and written in standard English?

Reviewer #1: Yes

Reviewer #2: No

Reviewer #3: No

Reviewer #1: 0.

The manuscript presents and discusses the practical applicability of CNN for the automated detection of post-operation(castration) pain of horses. The work presented involved the design of a shallow 2-conv-layer CNN for the classification of manually cropped images of horse faces, extracted from videos recorded before and after the castration surgery. The importance and relevance of the work are properly outlined in the introduction. The work reported a significant performance enhancement with the ensemble prediction using voting system, compared to individual CNN trained on a single face region of interest (ROI).

Nevertheless, the manuscript seems written in a hurry. Expression of sentences were not properly proof-read by the authors prior to submission.

Thank you for all the observations, they were very helpful during our editing process. 

The manuscript has now been reviewed by a native English speaker which we believe has helped improve the readability of the text.

For example, in Lines 101-108 there are duplication and badly organized phrases. Lines 133-137 are of different font than the other paragraphs.

Apologies, the text has been corrected.

However, there could still be valuable research contributions with following questions addressed, which have arised in the review process. In addition, the HGS-labelled image dataset generated in this study could potentially chip in to enhance the variation of publicly available HGS dataset for research.

The dataset generated in this study is available in the Mendeley data repository as we agree that the lack of datasets, that could be used for studying pain in animals is very limiting for robust studies. We do hope that these data will facilitate data sharing.

Question 1:

Lines 126-130:

How do 7 horses, for (2+4) days, for 4 times a day, result in 1252 videos?

(7 horses x 6 days x 4 times/day = 168 videos) Were multiple recording cameras used for every horse?

This information has been corrected in the manuscript (lines 128-130)

The actual number is 320 videos of 30 minutes duration each. (7 horses x 6 days x 4 times/day (more than a video per time point since the videos had a maximum length of 30 minutes). Thank you for pointing out our mistake.

Question 2:

It is commendable and helpful for reviewing that the authors have clearly provided the details like the CNN architecture, CNN kernel size, and the width and height of input images in Table 2 and Table 3.

Could the author explain why shallow CNN of only 2 convolutional layers were used? Couldn't a few additional convolutional layers benefit the feature extraction and abstraction capability of the model as the input are raw images and the output demanded is a highly abstract emotional expression of animal?

Indeed, we consider it important to show the configurations and final architecture so that the method can be replicated.

We agree that the more convolutional layers the better, but as convolutional layers are added, the number of input resources for subsequent layers is reduced and the processing demand increases significantly. Our goal is to show the method's viability, that's why we haven't advanced in testing other architectures. We hope that our methods can inspire future work that explores other architectures.

Question 3:

It is again commendable that authors honestly and clearly reported the low Recall performance for the presence of pain (esp. for the obvious pain) in each facial subcategory. This is important as the low recall indicates the failed detection of obvious pain which is of greater concern in pain management. This may reflect the insufficience of each individual facial category to serve in pain detection by the presented CNN model.

Has the author not considered increasing the reliability of HGS with independent labelling performed by more than only one trained-observer (as reported in line 159)?

Indeed, the analysis of the performance in each facial subcategory is relevant.

We have considered having more trained observers performing the labeling and we understand that this can improve the overall accuracy of the method. We have discussed this information in the text (Lines 304 – 307). However, we plan to carry out work, on this area, on for future research, since in this study our aim was to evaluate if this method would be useful. Now that we have this initial answer it is possible for us and other scientists to improve the details of the system, and perhaps exchange information and create solid partnerships.

Question 4:

For contents in Lines 202-207 & Lines 252-259:

Based on which image pool was the voting/ensemble model weight-tuned? Was it using the image pool that each member CNN of the ensemble model was trained on (as that presented in Table 1 & 3) or another separate image pool? Or was the weight-tuning also based on the pool of 30 images reported in Table 7 and Table 8 (which shouldn't be the case as the tuning process should not have access to the final test set)? The authors has not explicitly reported the image dataset used for the ensemble/voting weight-tuning process.

If this ensemble voting model with weight ratio of 3:1:1 for eye:ears:mouth&nostril is applied back onto the very original image dataset presented in Table 1 before these images are cropped into ROI, what is the performance of the model is terms of accuracy, precision and recall?

The weight tuning process (which is a key step resulting in the pain classification performance surge) should be analyzed and discussed in greater details.

What is shown in Table 1 is the number of cutouts made in original complete images. Automatic pain classification on the original image has not been implemented. This requires the development of another method (additional algorithm), in which the automatic search for the clippings of interest in the original image or video would be implemented, and then the proposed method would be applied. In our case, this process was performed manually, but for a final automatic system this development would be necessary.

Thank you for your observation, we agree that the voting model needed to be explained in more details. We have added explanation of the voting process, weight adjustments and image selection in the manuscript, (lines 207 – 216).

Reviewer #2: The paper aims at employing artificial intelligence in identifying the existence of pain and its level in horses. Although the problem addressed by the paper is exciting and carries great future potential in opening research avenues, the paper still needs much work to be publication worthy. I have the following concerns:

1. The writing of the paper needs revision. There are many repetitions that should have been omitted after the authors decided on the final shape of the paragraph. The first paragraph of the data collection section (starting line 101) is a prime example. These are not style mistakes but rather genuine writing/editing flaws. See line 58.

Thank you for your observations. These corrections were important and have been done. Also, the manuscript has now been reviewed by a native English speaker.

2. The authors need to enroll a computer science specialist with knowledge in AI. She/he will contribute to better experiments and to writing in the proper computer science jargon. This is a problem with the current manuscript as it is written by non-specialists.

The presented work is strongly multidisciplinary and our challenge was to create a report that could communicate the proposal and the results to scientists from different areas, such as computer science, veterinary and animal science. I would like to count on your understanding for this challenge since one of our aims is to show for other animal scientists and veterinarians how it is possible to have amazing outcomes when associating knowledge from the animal health area with computer science technology.

3. The data collection section, details of the drugs given are not necessary for the contribution of the paper (unless they are required for ethical approval).

Although the details of the drugs given may not be necessary from the perspective of the system development, we believe that this information could be relevant for other scientists working in the veterinary and animal science areas as it could interfere in the expression of pain.

4. Section III.II, the number of images 185.000 should be written as 185000, and there is no need for approximately as this is a simple count. The exact number can be easily included.

The information has been corrected in the manuscript (lines 141 – 143).

5. The authors need to elaborate more on the time and effort to inspect 185000 images to select 3000 of them (if not randomly). This does not seem right.

A computer program was used that, based on motion detection, selected frames in the videos. After this process, images with quality and representativeness of each pain level were manually selected, in order to select only the ones that made the pain evaluation possible. It was a very demanding task that took months of work, but we have managed to do it. We attempted in the text to clarify the information.

6. The approach to select the proper images could be better than just inspecting the individual frames, which carry a lot of repetitions especially at 30 frames per second video rate.

As answered in the previous question, in order not to have to inspect all the frames captured by the camera, we have used an application software that extracted frames only when motion was detected (We have added this information to the manuscript, line 141 -145). Then, the resulting frames were selected in order to get only the ones that were possible to evaluate pain through facial expressions. Of course, this is still a very demanding process, however we wanted to make sure that we got all the images possible in order to result in a good image database.

7. In general, the AI approach need to be driven by a computer scientist so that many of the issues in this paper could be ironed out. For example, the number of subjects (i.e., 7) is too small and may lead to overfitting of the model.

We agree that having more subjects would be ideal. Between these 7 seven subjects that we managed to include in this study, we had animals with different coat colors and different breeds, in order to improve the variability in the facial characteristics that the system would interpret. However, our proposal is novel and our goal with dissemination is to show its potential, and thus inspire other groups of scientists who can develop the work further. Right now, we have already been working on future developments for models with more subjects and a larger database. However, these ongoing improvements would benefit from the successful results that we would like to have feedback, from the current study.

8. The font size in paragraph on line 133 need to be consistent with the paper (such mistakes reflect badly on the quality of the paper).

It was corrected, thank you.

9. It is not clear how the system combines the result from the three body parts to form a judgement on pain. The current approach seems overly simple.

Thank you for your observation, we agree that the voting model needed to be explained in more details. We have added explanation of the voting process, weight adjustments and image selection in the manuscript, (lines 207 – 216).

10. Numbering of sections need to better.

We have reviewed the manuscript based on the PLOS ONE´s style requirements and corrected it.

Reviewer #3: The manuscript ‘Pain assessment in horses using automatic facial recognition through deep learning-based modeling’ is an interesting and timely manuscript. The results seem indeed promising to be able to record facial expressions of pain automatically and this is clearly worthwhile for practice.

However, I believe some aspects need to be addressed with more detail and there are a couple of things I do not understand which may need clarification first before the manuscript may be accepted for publication in PLOS ONE.

Below, you will find my arguments for the answers I have given to the questions that are part of the PLOS ONE review process and additional detailed feedback.

Title: I am not sure recognition is the right word to use. To me facial recognition would be that a computer can identify animal 1 from 2 or can tell this is an ear or a mouth. But in this study the assessment was aimed to say that if the ear was positioned like this, there is no pain present. That seems to be something different?

Thank you for these observations, they were very helpful to improve the manuscript.

We totally agree with this aspect related to the title. We have corrected the title for: Pain assessment in horses using automatic facial expression recognition through deep learning-based modeling.

Abstract

Line 2: ‘The aim of the study was ....’ instead of ‘The aim of the study is ....’?

The text was corrected.

Lines 6 (and 60): Yes, clear that training is important for using the HGS. However, a computer model needs to be trained too and if this is done not properly, it does not really help us forward. So not sure to what extend a computer model is an advantage then?

To create models based on supervised machine learning, training is required. However, after the construction of the final model (final algorithm), it can be used in any place indefinitely, being part of a computer vision system. On the other hand, when we are training people to apply the pain scale, we end up depending on their availability to evaluate the animal, which will not be the same as the system that could work full time, without supervision.

Lines 8 (and 64-66): True, but a person may not always be needed to be present? The animal can also be monitored from cameras for instance. So this is not always a problem for evaluation, I think.

We agree that a person's evaluation through a camera won't lead to a possible bias to the animal altering its pain expression due to a threatening stimulus. However, the idea is to create an automatic system that doesn't depend on human availability, since it would require large amounts of time to constantly evaluate each animal from time to time. The system would be able to do this task uninterruptedly.

Line 15: ‘process was applied’ and ‘methods were used’ instead of ‘.... is’ and ‘.... are’? May apply at other sentences too.

It was corrected.

Introduction

Line 28: [1,2] instead of [1][2]

It was corrected.

Lines 33 and 56: A space is missing before the reference. Please check whether a space is missing at other places perhaps too.

It was corrected.

Line 55: The word ‘if’’ is not needed, I think.

It was corrected.

Lines 59, 105 and 174: I think here is one space too many. Please check whether an extra space is present at other places perhaps too.

It was corrected.

Line 77: Full stop is missing after the references.

It was corrected.

Materials and Methods

Lines 88 and 92: I am not sure I understand what the authors mean with application exactly. To be able to reproduce the study, more details on this method may be needed to give.

It was complemented in the manuscript that “application” is referring to a “application software”. Line 86.

Lines 101-105: Please check these lines as things are said twice here.

It was corrected.

Line 125: What was the total length of the castration procedure? May be good to add to show the reader it was done as quickly as possible.

Thank you for your observation, we agree that this information is an important concern. The total length of the procedure was approximately 40 minutes (time considering anesthesia and recovery) including 20 minutes of the surgery procedure itself.

This information was added to the manuscript (lines 123 – 124).

Line 130: ‘videos of 30 minutes’ instead of ‘videos with 30 minutes?

It was corrected. (Line 129).

Line 130: With 1252 videos, this means you have about (1252 / 7 horses / 6 days / 4 time points =) 7 videos with a total of 3.5 hours of footage around a time point? That seems sufficient, I am just wondering how to visualize this so to say as there is for example only 2 hours in between time point 10 am and 12 pm. When was a video then considered to belong to a certain time point? Or do I not understand this correctly?

This information has been corrected in the manuscript (lines 128-130)

The actual number is 320 videos of 30 minutes duration each. (7 horses x 6 days x 4 times/day (more than a video per time point since the videos had a maximum duration of 30 minutes).

Lines 133-137: Seems to be in a different letter font?

It was corrected.

Lines 135 and 137: May be good to add when exactly on a day this was done, as I think it should not have interfered with making the videos of Line 130?

Post-castration follow ups were carried out from 07:00 am. Images were never collected when the presence of humans was recorded. It is possible that the images were collected and used for the study after the evaluations were carried out, however this should not have interfered with the measures obtained. Thank you for the useful question.

We have added this information to the manuscript (lines 134 – 136).

Lines 147-148: What is exactly part of the 4th and 5th parameter? You have to guess a bit where the 4th ends and the 5th begins.

We agree that this was not clear in the manuscript. We have corrected it (Lines 146 – 148).

Line 157/Fig. 2: I would like to suggest to take another picture as example for the mouth and nostrils parameter for moderately present. As I do not really understand what I am looking at here, I am afraid. If possible, a more clear one as used for not present and obviously present would definitely help.

In the picture of pain moderately present for the Mouth and nostrils parameter it is possible to notice that, in comparison to the pain not present image for the same parameter, the line between upper and lower lip (mouth column) is shorter, (but not as short as in the pain obviously present example). The mouth is slightly strained and so are the nostrils. However, we agree that this interpretation can be quite difficult, so we also added this information to the figure 2 legend in the manuscript. (Lines 157 – 161).

Lines 159-161: Was the pain assessment solely based on the HGS? Or also based on other behavioral or physiological parameters? In other words, how did the authors know the horse was really in pain? The horses were given post-operative analgesia, so perhaps they did not feel so much pain? For training and testing a computer model this may not be such an issue, but for practice it is. The ears, for instance, could also be turned backwards due to focusing on a sound and not related to pain, and I think that such false warnings to the veterinarian or horse owner is not desired. Or it is the case that only if all three parameters are scored as pain present by the model, a warning is given?

The reviewer's remarks are important. This is an initial work that seeks to show the potential of the proposed method. This method would only notify the presence of pain if there is more than one of the parameters indicating that state, therefore minimizing false warnings due to a change in only one of the parameters, for example the ears reacting to sound. The study was based solely on the published and validated HGS (Dalla Costa, 2014), and the question was related to the ability of a computer system to detect the changes. For a diagnosis to implement a treatment schedule, for example, threshold for responses could be used.

Lines 165-166: I am sorry, but I do not understand how the authors can have a final database of 4847 images if 3000 were selected in the first place? In addition, summing up 2379, 1436 and 1035, I get 4850 images. Somethings seems to be not entirely right here.

Each one of the original 3000 images of the horse´s full head has been used to generate image clippings for each one of the parameters, that is why it is possible to have more than 3000 images when adding up all the classes and parameters. We have corrected the mistake in the summing up and also added this explanation to the manuscript since we agree that this was not clear (Lines 169 – 172).

Line 173: ‘were built’ instead of ‘was built’

It was corrected.

Lines 177-181: I am not an expert on this yet, but what I learned is that others used much more images/frames for training (and testing) the model. Please see for instance: https://arxiv.org/pdf/1909.12605.pdf;
https://www.nature.com/articles/s41598-020-70688-6;
https://www.mdpi.com/1424-8220/19/5/1188/htm Perhaps it depends on what exactly the model needs to be trained in, but it may be good to add here why the authors know/feel these numbers are sufficient.

The reviewer's remarks are important. This is a preliminary work that seeks to show the potential of the proposed method. We have been working now on expanding the image bank to refine the computational model (algorithm), however, we needed to make sure that the system would be successful and promising in order to be able to gather resources for all the possible improvements. We also aim to spread the word to inspire other groups of scientists who can evolve with the work, and create potential collaborative arrangements.

Line 195/Table 2: Fig. 5 is not included in the manuscript?

Apologies, it was corrected. Where it was saying Fig. 5 is actually Table 3. (Line 200 – 203 – Table 2).

Lines 202-203: More details on the voting process would be interesting and relevant to give with respect to be able to reproduce the study.

Thank you for your observation, we agree that the voting model needed to be explained in more detail. We have added explanation of the voting process, weight adjustments and image selection in the manuscript, (lines 207 – 216).

Lines 205-206: ‘Thus, a variety of ... was tested’ instead of ‘Thus, it was tested a variety of .....’ makes more sense to me.

It was corrected.

Results and Discussion

Line 215: I would like to suggest to add ‘obviously’ to this line after ‘being’.

It was corrected.

Line 219: It took me some time to figure out where the numbers in Table 4 came from, until I realized that this was based on the test frames which was 237 images for the ears (10% of the total images). It would be good to repeat that somewhere here, I think. Please, be precise and clear which numbers are total and which used for training, validation and testing as the numbers given in Lines 214-215 are the overall total and not the ones used to get Table 4 and that may give a biased picture to the reader. May apply to Lines 231-233, 236-237 and 244-245 as well.

Thank you for this observation. This information was indeed not clear as it could be. We have added the details to precise the testing dataset used in the manuscript in lines (225 – 229) (241 – 244) (255 – 258).

In addition, are there not any results to mention for the validation part of the study?

Through the process of building models using artificial intelligence, databases are partitioned, with a fraction of the database being reserved for model building and validation and another part for the final performance evaluation. In our case representing 80%, 10% and 10% respectively.

Line 230: I would like to suggest to rephrase ‘on the eye’ as it feels as strange formulating in this way.

It was corrected. Line 238.

Lines 232 and 245; I would like to suggest to rephrase this as ‘moderately pain present, and xx of obviously pain present’, because that is more clear.

It was corrected.

Line 270: I would add ‘clearly’ or ‘obviously’ before ‘in pain’.

It was corrected.

Line 286: ‘in horses’ instead of ‘on horses’

It was corrected.

Line 289: Indeed a larger databank would be worthwhile. Do the authors have suggestions on the amount of images desired here?

There is no definitive answer, further research is needed. We are working to have a bank with 20000 images.

Line 293: Can the authors elaborate a bit more on what this bias would mean if present?

The aim is to minimize the impact of individual evaluation failures by using a larger number of evaluators. This information was added in the manuscript in order to make it more clear in lines (304 – 306).

Line 294: Nice to know the authors have in mind to include more behaviours. Do the authors have specific suggestions for these behaviours?

Our ongoing research at the veterinary teaching hospital has offered the opportunity to gather a large dataset on images obtained from clinical cases. Our goal is to explore further information on level of activity, eating, drinking, among other behaviors. Some behaviors that could also be important to be taken into account when evaluating pain are sleep, posture, facial expressions related to stress or aggression, posture, tail movement and others.

Conclusion

Line 303: Increasing the quality was not mentioned before, or was it? If not, I believe one cannot raise new aspects in the conclusion, so please include the quality in the discussion first as well.

We have included this aspect in the discussion too in line 301.

---

## [Decision Letter · Decision Letter 1]

16 Jul 2021

PONE-D-21-11680R1

Pain assessment in horses using automatic facial expression recognition through deep learning-based modeling

PLOS ONE

Dear Dr. Lencioni,

Thank you for submitting your manuscript to PLOS ONE. After careful consideration, we feel that it has merit but does not fully meet PLOS ONE’s publication criteria as it currently stands. Therefore, we invite you to submit a revised version of the manuscript that addresses the points raised during the review process.

We look forward to receiving your revised manuscript.

Kind regards,

Humaira Nisar

Academic Editor

PLOS ONE

Additional Editor Comments:

Based on the reviewers comments, the manuscript is not ready for publication yet.

Reviewers' comments:

Reviewer's Responses to Questions

**Comments to the Author**

1. If the authors have adequately addressed your comments raised in a previous round of review and you feel that this manuscript is now acceptable for publication, you may indicate that here to bypass the “Comments to the Author” section, enter your conflict of interest statement in the “Confidential to Editor” section, and submit your "Accept" recommendation.

Reviewer #1: (No Response)

Reviewer #3: (No Response)

2. Is the manuscript technically sound, and do the data support the conclusions?

Reviewer #1: Partly

Reviewer #3: Yes

3. Has the statistical analysis been performed appropriately and rigorously? 

Reviewer #1: N/A

Reviewer #3: N/A

4. Have the authors made all data underlying the findings in their manuscript fully available?

Reviewer #1: Yes

Reviewer #3: Yes

5. Is the manuscript presented in an intelligible fashion and written in standard English?

Reviewer #1: Yes

Reviewer #3: Yes

6. Review Comments to the Author

Reviewer #1: 【Question 1 follow-up】:

Authors' response is accepted. Corrective action has been made.

【Question 2 & 3 follow-up】:

Authors have responded by saying further future work will be done in these directions.

It is okay for Question 3 not to be addressed with corrective research action as it will require almost another full round of research work apart from the video recording.

【Question 4 follow-up】:

According to Table 1, the datasets are highly imbalanced (within the 3 classes: absence of pain, moderate pain, and obvious pain) with vast majority of the images being under absence of pain. So, the focus of performance analysis would naturally be put on the Recall and Precision of the categories with pain (moderate pain and obvious pain). After all, this model is for the purpose of pain assessment.

Using only one of the three facial regions (the ears, eyes, or the nostrils), the system's pain detection Recall performance was from around 27%-67% as recorded in Tables 4, 5, and 6. This high-accuracy low-recall performance could be due to the vastly imbalanced traininng image pool, with the CNN parameters being tuned to favor the detection of the absence of pain. Perhaps the authors could consider improving the balance of the dataset, or providing counter-measure for this issue.

The balance of the dataset is important especially for the test dataset. Out of the 237 test images in the ear category, 198 of them are under the absence of pain. The model performance tested using such an imbalanced test set is likely not properly reflecting the true accuracy of the model.

It was reported that only 10 random images (out of the total of 3000 selected images) were used for the performance testing of the final ensemble voting model. Are there any valid reasons for why the authors would not allocate a greater amount of images for the final performance test? Just for instance, at a ratio of 6:1:1:1:1 for the following purposes:

- 60% (for individual CNN training):

- 10% (for individual CNN validation):

- 10% (for individual CNN testing):

- 10% (for ensemble voting weight tuning):

- 10% (for final ensemble test)

, instead of using only the same 10 images out of the 3000 for both the voting-weight tuning and final testing.

Furthermore,

【Page 12. Lines 223-227】

The same images were used for tuning the "voting-weight" ratio and for the final ensembled performance test. This is not a correct practice. By doing so (using the same images for tuning and testing), the reported weight ratio was not properly validated and may not be well applicable onto wider variations of pain expression.

As the authors have responded to my Question 2 & 3, saying that "in this study our aim was to evaluate if this method would be useful" and "hope that our methods can inspire future work that explores other architectures", the main concern may not be on reporting very high classification accuracy. However, it is important to present an analysis report that is technically and statistically sound.

【Additionally】,

For facilitating the review process of the contribution of this paper, it would be helpful to include a systematic literature review on others' previously-published relevant research and their relevant research achievement, in order to present a clearer overall picture of current state of research in this subject of interest. For instance, the automated decoding of expression or pain in horses. Or the automated decoding of expression or pain in other animals, if the reported works on horses are currently too limited.

Reviewer #3: I would like to thank the authors for their effort to revise the manuscript according to the reviewers’ suggestions, and generally I am satisfied with the revisions made. I do have, however, some last suggestions and questions. If the authors revise the manuscript accordingly, I will accept the manuscript for publication.

Last suggestions and questions:

L5: Scale with a capital S instead of a small s?

L64: Either ‘a potential threatening stimulus’ or ‘potential threatening stimuli’

L98: Images with a small i instead of a capital I?

L101: space is missing between was and requested. In L103 is a space missing as well.

L129: I would say ‘in 320 videos of 30 minutes each’ instead of ‘in 320 videos each of which was 30 minutes in duration’.

L135: I would like to thank the authors for including the time here. However, the time is equal to the first observation point of the day. I assume that the post-operative procedures were done first and video recordings started thereafter? In addition, the penicillin-based ointment was applied twice a day. When was the second time?

L157/Fig. 2: Although the extra explanation in the figure caption helps, I still find it difficult to see where the mouth and nostrils are in the picture of this parameter in the moderately pain option. I mean for not present and obviously present the nostrils are clearly on the right-side and the mouth on the left, but this is very vague for the in between pain option. Also, in the other two pictures, there is clearly some background beneath the mouth making it clearly standing out from the background/other body parts, but this seems not the case in the picture I have concerns with. If the authors would have a different and clearer picture, that would be great. Otherwise it might work to give even more explanation in the caption regarding e.g. whether the nostrils are on the left or right for instance.

L170: all instead of both?

L253-254: Here ear and eye in singular, but earlier in plural. Good to make this consistent?

L299: Perhaps add here to the manuscript and not only in reply to the reviewers some words about that this work is “preliminary work that seeks to show the potential of the proposed method. [...] We also aim to spread the word to inspire other groups of scientists who can evolve with the work, and create potential collaborative arrangements.”. In this way the authors show that they are, for instance, aware of the model’s shortcomings at this stage in the process. Thus, that it is a work in process.

7. PLOS authors have the option to publish the peer review history of their article (what does this mean?). If published, this will include your full peer review and any attached files.

Reviewer #1: No

Reviewer #3: No

---

## [Author Response · Author response to Decision Letter 1]

28 Aug 2021

Dear Dr. Humainara Nisar and reviewers, 

Thank you very much for your comments, we have addressed all the points with extreme attention and we believe that now the manuscript has improved even more. We would like to thank you again for such important considerations.

In the document below we are describing how we answered the considerations and changes on the manuscript.

Additional Editor Comments:

Based on the reviewers comments, the manuscript is not ready for publication yet.

Reviewers' comments:

Reviewer's Responses to Questions

Comments to the Author

1. If the authors have adequately addressed your comments raised in a previous round of review and you feel that this manuscript is now acceptable for publication, you may indicate that here to bypass the “Comments to the Author” section, enter your conflict of interest statement in the “Confidential to Editor” section, and submit your "Accept" recommendation.

Reviewer #1: (No Response)

Reviewer #3: (No Response)

2. Is the manuscript technically sound, and do the data support the conclusions?

Reviewer #1: Partly

Reviewer #3: Yes

3. Has the statistical analysis been performed appropriately and rigorously?

Reviewer #1: N/A

Reviewer #3: N/A

4. Have the authors made all data underlying the findings in their manuscript fully available?

Reviewer #1: Yes

Reviewer #3: Yes

5. Is the manuscript presented in an intelligible fashion and written in standard English?

Reviewer #1: Yes

Reviewer #3: Yes

6. Review Comments to the Author

Reviewer #1: 【Question 1 follow-up】:

Authors' response is accepted. Corrective action has been made.

【Question 2 & 3 follow-up】:

Authors have responded by saying further future work will be done in these directions.

It is okay for Question 3 not to be addressed with corrective research action as it will require almost another full round of research work apart from the video recording.

Thank you for understanding the limitations to address question 3. We are committed to continue this area or research and your comments are certainly valuable to improve our future work.

【Question 4 follow-up】:

According to Table 1, the datasets are highly imbalanced (within the 3 classes: absence of pain, moderate pain, and obvious pain) with vast majority of the images being under absence of pain. So, the focus of performance analysis would naturally be put on the Recall and Precision of the categories with pain (moderate pain and obvious pain). After all, this model is for the purpose of pain assessment.

Using only one of the three facial regions (the ears, eyes, or the nostrils), the system's pain detection Recall performance was from around 27%-67% as recorded in Tables 4, 5, and 6. This high-accuracy low-recall performance could be due to the vastly imbalanced traininng image pool, with the CNN parameters being tuned to favor the detection of the absence of pain. Perhaps the authors could consider improving the balance of the dataset, or providing counter-measure for this issue.

The balance of the dataset is important especially for the test dataset. Out of the 237 test images in the ear category, 198 of them are under the absence of pain. The model performance tested using such an imbalanced test set is likely not properly reflecting the true accuracy of the model.

It was reported that only 10 random images (out of the total of 3000 selected images) were used for the performance testing of the final ensemble voting model. Are there any valid reasons for why the authors would not allocate a greater amount of images for the final performance test? Just for instance, at a ratio of 6:1:1:1:1 for the following purposes:

- 60% (for individual CNN training):

- 10% (for individual CNN validation):

- 10% (for individual CNN testing):

- 10% (for ensemble voting weight tuning):

- 10% (for final ensemble test)

, instead of using only the same 10 images out of the 3000 for both the voting-weight tuning and final testing.

Thank you for all your relevant considerations, they were really helpful in order to improve the quality of the manuscript. We agree that ideally the classes should have a more balanced design aiming to an even better and more robust outcome for the reported system. We discussed in the manuscript that working with images of animals, especially in pain models, brings the associated challenges to work with small datasets. That is why we chose to use all the images possible for each model, even though this would result in an unbalanced model the training for each was the best that we were able to do. We are developing now on our current research, ways to improve the dataset and as a result we should be able to improve the system with a more balanced model adding more images. 

We have made changes in our final algorithm to address the issues raised regarding the number of images and the performance test, which we will discuss in detail in the next question, since it is also related to the tuning process.

Furthermore,

【Page 12. Lines 223-227】

The same images were used for tuning the "voting-weight" ratio and for the final ensembled performance test. This is not a correct practice. By doing so (using the same images for tuning and testing), the reported weight ratio was not properly validated and may not be well applicable onto wider variations of pain expression.

As the authors have responded to my Question 2 & 3, saying that "in this study our aim was to evaluate if this method would be useful" and "hope that our methods can inspire future work that explores other architectures", the main concern may not be on reporting very high classification accuracy. However, it is important to present an analysis report that is technically and statistically sound.

We would like to thank the reviewer for the important comments. We were able to make improvements to our system and we believe that now our results are more robust and accurate. As a solution for these concerns, we chose to substitute the voting system for a machine learning method, based on Artificial Neural Network (ANN-based classifier) and a Perceptron feedforward and multi-layered architecture, with a sigmoid transfer function in the hidden layer and a linear transfer function in the output layer. The Levenberg-Marquardt backpropagation method and mean squared error were employed to measure the performance using k-fold cross validation of 10. In addition to that, we have improved our database for the process of training and testing of the classifier, that went from a total of 30 images to 120 images on the new model. With these changes, the overall accuracy of the final model was slightly lower than the previous, however we believe that it was worth in order to address all the topics discussed previously. Again, thank you for your comments 

【Additionally】,

For facilitating the review process of the contribution of this paper, it would be helpful to include a systematic literature review on others' previously-published relevant research and their relevant research achievement, in order to present a clearer overall picture of current state of research in this subject of interest. For instance, the automated decoding of expression or pain in horses. Or the automated decoding of expression or pain in other animals, if the reported works on horses are currently too limited.

Recently we came across the article (Andersen PH, Broomé S, Rashid M, Lundblad J, Ask K, Li Z, et al. Towards machine recognition of facial expressions of pain in horses. Animals. 2021;11. doi:10.3390/ani11061643) that offers a comprehensive review on the complexities of assessing pain in horses using automatic methods. We have added this article in the manuscript to present a clearer overall picture of this research topic.

Reviewer #3: I would like to thank the authors for their effort to revise the manuscript according to the reviewers’ suggestions, and generally I am satisfied with the revisions made. I do have, however, some last suggestions and questions. If the authors revise the manuscript accordingly, I will accept the manuscript for publication.

We are very happy to hear that and we would like to thank you for the previous and current revisions, that were extremely helpful to improve the manuscript.

Last suggestions and questions:

L5: Scale with a capital S instead of a small s?

We have corrected it in the Manuscript.

L64: Either ‘a potential threatening stimulus’ or ‘potential threatening stimuli’

We have corrected it in the Manuscript.

L98: Images with a small i instead of a capital I?

We have corrected it in the Manuscript.

L101: space is missing between was and requested. In L103 is a space missing as well.

We have corrected it in the Manuscript.

L129: I would say ‘in 320 videos of 30 minutes each’ instead of ‘in 320 videos each of which was 30 minutes in duration’.

We have corrected it in the Manuscript.

L135: I would like to thank the authors for including the time here. However, the time is equal to the first observation point of the day. I assume that the post-operative procedures were done first and video recordings started thereafter? In addition, the penicillin-based ointment was applied twice a day. When was the second time?

We would like to thank you for this observation. This information was not clear in the manuscript, so we added the observation that the follow ups were carried out before the video recordings in the morning (7am) and in the afternoon (12 pm).

L157/Fig. 2: Although the extra explanation in the figure caption helps, I still find it difficult to see where the mouth and nostrils are in the picture of this parameter in the moderately pain option. I mean for not present and obviously present the nostrils are clearly on the right-side and the mouth on the left, but this is very vague for the in between pain option. Also, in the other two pictures, there is clearly some background beneath the mouth making it clearly standing out from the background/other body parts, but this seems not the case in the picture I have concerns with. If the authors would have a different and clearer picture, that would be great. Otherwise it might work to give even more explanation in the caption regarding e.g. whether the nostrils are on the left or right for instance.

Thank you for this observation. We have changed the image and believe that it is better to understand all the information now.

L170: all instead of both?

We have corrected it in the Manuscript.

L253-254: Here ear and eye in singular, but earlier in plural. Good to make this consistent?

We have corrected it in the Manuscript.

L299: Perhaps add here to the manuscript and not only in reply to the reviewers some words about that this work is “preliminary work that seeks to show the potential of the proposed method. [...] We also aim to spread the word to inspire other groups of scientists who can evolve with the work, and create potential collaborative arrangements.”. In this way the authors show that they are, for instance, aware of the model’s shortcomings at this stage in the process. Thus, that it is a work in process.

Thank you for this observation, we have added this information to the manuscript.

7. PLOS authors have the option to publish the peer review history of their article (what does this mean?). If published, this will include your full peer review and any attached files.

Do you want your identity to be public for this peer review? For information about this choice, including consent withdrawal, please see our Privacy Policy.

Reviewer #1: No

Reviewer #3: No

---

## [Decision Letter · Decision Letter 2]

15 Sep 2021

PONE-D-21-11680R2Pain assessment in horses using automatic facial expression recognition through deep learning-based modelingPLOS ONE

Dear Dr. Lencioni,

Thank you for submitting your manuscript to PLOS ONE. After careful consideration, we feel that it has merit but does not fully meet PLOS ONE’s publication criteria as it currently stands. Therefore, we invite you to submit a revised version of the manuscript that addresses the points raised during the review process.

We look forward to receiving your revised manuscript.

Kind regards,

Humaira Nisar

Academic Editor

PLOS ONE

Journal Requirements:

Additional Editor Comments (if provided):

Reviewers' comments:

Reviewer's Responses to Questions

**Comments to the Author**

1. If the authors have adequately addressed your comments raised in a previous round of review and you feel that this manuscript is now acceptable for publication, you may indicate that here to bypass the “Comments to the Author” section, enter your conflict of interest statement in the “Confidential to Editor” section, and submit your "Accept" recommendation.

Reviewer #1: (No Response)

Reviewer #3: All comments have been addressed

2. Is the manuscript technically sound, and do the data support the conclusions?

Reviewer #1: Partly

Reviewer #3: (No Response)

3. Has the statistical analysis been performed appropriately and rigorously? 

Reviewer #1: N/A

Reviewer #3: (No Response)

4. Have the authors made all data underlying the findings in their manuscript fully available?

Reviewer #1: Yes

Reviewer #3: (No Response)

5. Is the manuscript presented in an intelligible fashion and written in standard English?

Reviewer #1: Yes

Reviewer #3: (No Response)

6. Review Comments to the Author

Reviewer #1: I think the revised manuscript is acceptable for publication.

Thanks to the authors for their passionate research contribution and professional effort at revising the manuscript in order to address the reviewers' comments.

Nevertheless, I have 3 comments (a minor grammatical comment and two other technical comments) as below:

Comment 1:

【line 220 in manuscript (or line 225 in manuscript with tracked changes)】

- resulting in (instead of resulting on)

Comment 2:

【lines 272-274 in manuscript】

"The final value for the number of neurons in the hidden layer, the learning rate and the momentum hyperparameters were 5, 0.3 and 0.2 respectively."

If it is possible, it could be valuable to report the final weights and biases of the 5 hidden nodes, since the weights and biases of these 5 hidden nodes carry information regarding the relative significance of 3 facial regions and the number of nodes (3 x 5 + 5 = 20 values ?) are not too large too be presentable.

Comment 3 (quite an important query as to the validity of the 10-fold cross validation results):

- Referring to 【lines 217-219】 & 【lines 223-225】 of the manuscript with tracked changes or 【lines 216-218】 & 【lines 219-221】 of the manuscript,

"The Levenberg-Marquardt backpropagation method and mean squared error were employed to measure the performance using k-fold cross validation of 10" and

"Forty complete original images were randomly selected of animals from each class: no pain, moderately present pain and obviously present pain, resulting on 120 complete images."

But the results reported in Table 7 and Table 8 contained all the 120 images and this is not the averaged results of the 10-fold cross validation.

May I know how the 10-fold validation was conducted and why the results reported is not the averaged 10-fold cross-validated result?

Using 120 images for 10-fold cross validation, each fold of validation should have only 12 validation images, not the 120 as reported in Tables 7 & 8. Or has the author presented the summation of the 10-fold cross-validation, instead of the average? This should be clarified in the Table's caption or the content of manuscript to avoid confusion.

End of Recommendation

Reviewer #3: (No Response)

7. PLOS authors have the option to publish the peer review history of their article (what does this mean?). If published, this will include your full peer review and any attached files.

Reviewer #1: No

Reviewer #3: No

---

## [Author Response · Author response to Decision Letter 2]

21 Sep 2021

Dear Dr. Humainara Nisar and reviewers, 

Thank you very much for your important considerations, we have addressed all of them and we hope that the manuscript is now acceptable for publication.

In the document below we are describing how we answered the considerations and carried out the suggested changes on the manuscript.

Journal Requirements:

We reviewed the reference list and we understand that it is complete and correct.

Additional Editor Comments (if provided):

Reviewers' comments:

Reviewer's Responses to Questions

Comments to the Author

1. If the authors have adequately addressed your comments raised in a previous round of review and you feel that this manuscript is now acceptable for publication, you may indicate that here to bypass the “Comments to the Author” section, enter your conflict of interest statement in the “Confidential to Editor” section, and submit your "Accept" recommendation.

Reviewer #1: (No Response)

Reviewer #3: All comments have been addressed

2. Is the manuscript technically sound, and do the data support the conclusions?

Reviewer #1: Partly

Reviewer #3: (No Response)

3. Has the statistical analysis been performed appropriately and rigorously?

Reviewer #1: N/A

Reviewer #3: (No Response)

4. Have the authors made all data underlying the findings in their manuscript fully available?

Reviewer #1: Yes

Reviewer #3: (No Response)

5. Is the manuscript presented in an intelligible fashion and written in standard English?

Reviewer #1: Yes

Reviewer #3: (No Response)

6. Review Comments to the Author

Reviewer #1: I think the revised manuscript is acceptable for publication.

Thanks to the authors for their passionate research contribution and professional effort at revising the manuscript in order to address the reviewers' comments.

Nevertheless, I have 3 comments (a minor grammatical comment and two other technical comments) as below:

Comment 1:

【line 220 in manuscript (or line 225 in manuscript with tracked changes)】

- resulting in (instead of resulting on)

Thank you for your comment. It was corrected in the manuscript.

Comment 2:

【lines 272-274 in manuscript】

"The final value for the number of neurons in the hidden layer, the learning rate and the momentum hyperparameters were 5, 0.3 and 0.2 respectively."

If it is possible, it could be valuable to report the final weights and biases of the 5 hidden nodes, since the weights and biases of these 5 hidden nodes carry information regarding the relative significance of 3 facial regions and the number of nodes (3 x 5 + 5 = 20 values ?) are not too large too be presentable.

Thank you for your comment.

All generated models and the buffer results were added to the data repository. 

Lencioni, Gabriel; Sousa, Rafael; Sardinha, Edson; Romero, Rodrigo; Zanella, Adroaldo (2021), “Automatic Pain Assessment in Horses”, Mendeley Data, V3, doi: 10.17632/t8rtzcgwxm.3

Comment 3 (quite an important query as to the validity of the 10-fold cross validation results):

- Referring to 【lines 217-219】 & 【lines 223-225】 of the manuscript with tracked changes or 【lines 216-218】 & 【lines 219-221】 of the manuscript,

"The Levenberg-Marquardt backpropagation method and mean squared error were employed to measure the performance using k-fold cross validation of 10" and

"Forty complete original images were randomly selected of animals from each class: no pain, moderately present pain and obviously present pain, resulting on 120 complete images."

But the results reported in Table 7 and Table 8 contained all the 120 images and this is not the averaged results of the 10-fold cross validation.

May I know how the 10-fold validation was conducted and why the results reported is not the averaged 10-fold cross-validated result?

Using 120 images for 10-fold cross validation, each fold of validation should have only 12 validation images, not the 120 as reported in Tables 7 & 8. Or has the author presented the summation of the 10-fold cross-validation, instead of the average? This should be clarified in the Table's caption or the content of manuscript to avoid confusion.

Thank you for your comment. We carried out the suggested changes in order to address these issues. We believe that the title of table 7 and 8 could lead to this confusion. We have changed the title of both tables and explained in the methods section that the performance metrics were based on the average values of all the folds. (The performance indicators are calculated by the average values and the table showing the hits and errors is formed by the summation: https://www.cs.waikato.ac.nz/ml/weka/mooc/dataminingwithweka/transcripts/Transcript2-5.txt).

End of Recommendation

Reviewer #3: (No Response)

7. PLOS authors have the option to publish the peer review history of their article (what does this mean?). If published, this will include your full peer review and any attached files.

Do you want your identity to be public for this peer review? For information about this choice, including consent withdrawal, please see our Privacy Policy.

Reviewer #1: No

Reviewer #3: No

---

## [Decision Letter · Decision Letter 3]

4 Oct 2021

Pain assessment in horses using automatic facial expression recognition through deep learning-based modeling

PONE-D-21-11680R3

Dear Dr. Lencioni,

We’re pleased to inform you that your manuscript has been judged scientifically suitable for publication and will be formally accepted for publication once it meets all outstanding technical requirements.

Kind regards,

Humaira Nisar

Academic Editor

PLOS ONE

Additional Editor Comments (optional):

Thank you very much for responding to the reviewer comments which has resulted in significant improvement. The manuscript is now ready for publication. Congratulations.

Reviewers' comments:

Reviewer's Responses to Questions

**Comments to the Author**

1. If the authors have adequately addressed your comments raised in a previous round of review and you feel that this manuscript is now acceptable for publication, you may indicate that here to bypass the “Comments to the Author” section, enter your conflict of interest statement in the “Confidential to Editor” section, and submit your "Accept" recommendation.

Reviewer #1: All comments have been addressed

2. Is the manuscript technically sound, and do the data support the conclusions?

Reviewer #1: Yes

3. Has the statistical analysis been performed appropriately and rigorously? 

Reviewer #1: N/A

4. Have the authors made all data underlying the findings in their manuscript fully available?

Reviewer #1: Yes

5. Is the manuscript presented in an intelligible fashion and written in standard English?

Reviewer #1: Yes

6. Review Comments to the Author

Reviewer #1: (No Response)

7. PLOS authors have the option to publish the peer review history of their article (what does this mean?). If published, this will include your full peer review and any attached files.

Reviewer #1: No

---

## [Editor Report · Acceptance letter]

11 Oct 2021

PONE-D-21-11680R3 

Pain assessment in horses using automatic facial expression recognition through deep learning-based modeling 

Dear Dr. Lencioni:

I'm pleased to inform you that your manuscript has been deemed suitable for publication in PLOS ONE. Congratulations! Your manuscript is now with our production department. 

Kind regards, 

on behalf of

Dr. Humaira Nisar 

Academic Editor

PLOS ONE